

# Sensitivity of woody carbon stocks to bark investment strategy in Neotropical savannas and forests

Anna T. Trugman[1,2], David Medvigy[1,3], William .A. Hoffmann[4], Adam F.A. Pellegrini[5]

[1]Program in Atmospheric and Oceanic Sciences, Princeton University, Princeton, NJ 08544, USA

[2]Department of Biology, University of Utah, Salt Lake City, UT 84112, USA

[3]Department of Biological Sciences, University of Notre Dame, Notre Dame, IN 46556, USA

[4]Department of Plant Biology, North Carolina State University, Raleigh, NC 27695, USA

[5]Department of Earth System Science, Stanford University, Stanford, CA 94305, USA

Correspondence to: A.T. Trugman (a.trugman@utah.edu) and A.F.A. Pellegrini (afapelle@stanford.edu)



**Abstract.** Fire frequencies are changing in Neotropical savannas and forests as a result of forest fragmentation and increasing drought. Such changes in fire regime and climate are hypothesized to decrease the stability of tropical carbon storage, but there has been little consideration of the widespread variability in tree fire tolerance strategies. To test how aboveground carbon stocks change with fire frequency and community composition, we update the ED2 model with (i) a fire survivorship module based on tree bark thickness (a key fire-tolerance trait across woody plants in savannas and forests), and

(ii) plant functional types representative of trees in the region. With these updates, the model is better able to predict how fire frequency affects population demography and aboveground woody carbon. Simulations illustrate that the high survival rate of thick-barked, large trees reduces carbon losses with increasing fire frequency, with high investment in bark being particularly important in reducing losses in the wettest sites. Additionally, in landscapes that frequently burn, bark investment can broaden the range of climate and fire conditions under which savannas occur by reducing the range of

conditions leading to either complete tree loss or complete grass loss. These results highlight that woody biomass carbon stocks in the tropics depend not only on changing fire frequencies, but also on tree fire survival strategy. Incorporation of a bark investment strategy in vegetation models holds promise for improving predictions of landscape-level carbon dynamics and savanna distribution, particularly in the context of global climate change.



## 1 Introduction


Tropical savannas and forests are important components of the land carbon sink (Pan et al., 2011; Liu et al., 2015; Ahlström et al., 2015). However, their ability to continue sequestering carbon is uncertain (Malhi et al., 2008), due in part to the impact of projected increases in drought frequency and changes in fire regime on woody carbon stocks (Brando et al., 2014). Globally, tropical forests, savannas, and grasslands comprise ~60% of total terrestrial gross primary productivity

(Beer et al., 2010), but are also responsible for over 65% of global carbon emissions stemming from fire and deforestation (van der Werf et al., 2010; van der Werf et al., 2009). The balance between biological carbon uptake through photosynthesis and carbon emissions from respiration and biomass burning determine whether tropical savannas and forests are a net carbon sink or source.

Fire is critical in defining the vegetation structure and distribution of tropical savannas and forests (Bond et al.,

2005; Hoffmann et al., 2012a; Staver et al., 2011b). A positive feedback between flammable grass presence and tree fire mortality in open canopies has been identified as an important mechanism in maintaining savanna regions (Archibald et al., 2009; Hoffmann et al., 2012b; Staver et al., 2011a). Some trees can persist, despite high fire frequencies in many savannas, because they invest heavily in building thick bark (Pellegrini et al., 2017a). Thick bark insulates the xylem and phloem from fire damage, increasing the probability of tree fire survival (Brando et al., 2012; Hoffmann et al., 2012a), and potentially

decreasing ecosystem carbon vulnerability to increasing fire frequency with global climate change (Pellegrini et al., 2016b). However, this increase in bark investment comes at a growth cost: thicker-barked savanna species grow more slowly than thinner-barked forest species under similar growing conditions (Hoffmann et al., 2012a; Rossatto et al., 2009). Though many savanna trees have growth strategies that employ thick bark as a fire survival mechanism, species vary greatly in their investment in bark (Pausas, 2015; Pellegrini et al., 2017a; Rosell, 2016).


Climate, particularly precipitation, has the potential to interact with fire frequency and tree growth strategy (Brando et al., 2014). In locations with low precipitation, tree growth rates are much slower than in locations with high precipitation (Baker et al., 2003). Slower growth rates result in a population of smaller trees with relatively thinner bark than their larger counterparts, making it more difficult for trees to grow and survive in high fire frequency regimes in dry regions. In addition to climate, physiology can play a role because some trees grow more rapidly than others due to differences in maximum

photosynthetic capacity or specific leaf area (Rossatto et al., 2009). Tree growth rate is critical for determining the ability of trees to recover from fire, and consequently the fire frequency necessary to maintain savanna.

Simulating savanna vegetation dynamics is notoriously difficult. However, progress has recently been made in identifying some key mechanisms needed to stabilize savannas in dynamic global vegetation models (DGVMs) (Scheiter and Higgins, 2009; Lehsten et al., 2016; Baudena et al., 2015; Higgins et al., 2000; Haverd et al., 2013; Lasslop et al., 2016).

Despite recent progress, DVGMs are still unable to fully capture global savanna extent as emergent features, generally over-





predicting the scope of either grasslands or tropical forests (Lasslop et al., 2016). This makes it difficult to quantify the impacts of projected climate change on tropical carbon storage because both carbon storage capacity and the resistance of ecosystem carbon to changes in precipitation and fire regime vary across tropical biomes. Additional mechanisms observed to be important for maintaining tree-grass coexistence in empirical studies, such as variability in tree fire survival strategy

(Hoffmann et al., 2012a), are underrepresented in models (Haverd et al., 2013; Lehsten et al., 2016). As a result, the implications of including variability in tree fire survival strategy are relatively untested, providing one useful avenue to improve modelling of savanna-forest dynamics.

To better understand the sensitivity of tropical carbon storage to changes in rainfall regime and fire frequency, we updated the Ecosystem Demography model 2 (ED2) to include distinct tropical savanna and forest plant functional types

(PFTs), each with a different bark investment strategy. We evaluated our model's predictions against observations of savanna and forest tree growth rates, tree inventories, and total aboveground carbon (AGB) for different fire frequencies using field data from savannas and forests in the Cerrado region of Brazil. We then tested whether two hypotheses were consistent with model simulations. (1) Including bark investment as a tree fire survival strategy decreases simulated carbon losses to increasing fire frequency, regardless of precipitation regime, due to the higher probability of survival for thicker-

barked, larger trees. (2) Including bark thickness as a fire survival strategy expands the environmental conditions under which trees and grasses coexist by allowing for increased tree survival in frequently burned savannas. Finally, we considered the effects of changing fire frequency and fire survival strategy on ecosystem composition and aboveground carbon vulnerability along a rainfall gradient in the Neotropics.

**2 Materials and Methods**

**2.1 Model Description**

Our model simulations were carried out in a cohort-based terrestrial biosphere model, ED2. ED2 explicitly scales up tree-level competition for light, water, and nutrients to the ecosystem level (Medvigy et al., 2009; Medvigy and Moorcroft, 2012). The effects of water limitation on photosynthesis have been previously identified to be important for

simulating savanna-grass dynamics (Baudena et al., 2015; Lasslop et al., 2016). Correspondingly, a novel aspect of the version of ED2 used in this study is the mechanistic representation of water-limited photosynthesis for $C_3$ plants, whereby leaf and stem water potential are tracked and used to solve for root zone water uptake, transport of water vertically through the sapwood, and transpiration of water into the atmosphere. Variability in hydraulic traits such as turgor loss point, xylem water conductivity, and marginal water use efficiency determine PFT-specific responses to changes in leaf and stem water

potential. Importantly, this mechanistic water limitation scheme has been demonstrated to better resolve vegetation dynamics in water limited tropical ecosystems (Xu et al., 2016), such as tropical savanna and forest regions.



We have incorporated in ED2 the following new processes important to ecosystem fire resistance in the Neotropics: (1) a PFT-specific bark investment strategy in two updated tropical PFTs; (2) a carbon tradeoff between bark production and tree height, canopy area, sapwood area, rooting depth, and leaf carbon; (3) a fire survivorship function dependent on

individual tree bark thickness; and (4) a dynamic feedback between tree size, survivorship probability, and grass biomass availability. Updated model codes are included as Supplementary Materials file S1. The two new PFTs represent a generic tropical forest tree PFT and tropical savanna tree PFT. Both are based on the tropical brevideciduous PFT from Xu et al. (2016). Previous PFT versions do not include a bark thickness trait. The brevideciduous PFT was chosen because its intermediate wood density and specific leaf area represent a drought survival strategy incorporating both drought avoidance

and resistance. This intermediate strategy is utilized by a broad array of tropical tree species (Xu et al., 2016). The savanna and forest PFTs differ only in their bark investment strategy and the associated trade-offs. In the model, individual tree bark thickness is calculated according to the following equation (Thonicke et al., 2010):

$$bt(PFT, dbh) = \beta(PFT) * dbh \ . \tag{1}$$


In Equation (1), $bt$ is tree bark thickness in cm, $\beta$ is the bark thickness slope coefficient (Table 1) that varies with PFT, and $dbh$ is tree diameter at breast height in cm. In Neotropical savannas and forests, $\beta$ ranges from ~0.068- 0.087 for trees of the forest functional guild and from ~0.087-0.142 for trees of the savanna functional guild (Pellegrini et al., 2016a). We assigned $\beta = 0.077$ and $\beta = 0.110$ to our forest and savanna PFTs, respectively, based on the average of the observed ranges in bark

thickness.

The carbon cost of a tree investing in bark is difficult to quantify. Here, we incorporated a cost through the tree allometric relations. Bark turnover rate in these systems is assumed to be negligible. Ordinarily in cohort- or individual-based models, dbh is allometrically related to woody biomass, leaf biomass, crown height, crown area, rooting depth, and sapwood area. The model also uses a standard allometric relationship between dbh and woody biomass. However, the new

model relates the other derived properties to dbh with bark excluded (denoted dbh´). Thus, for a given dbh, a PFT with a large β will have a smaller dbh´, fewer leaves, a shorter height, a shallower rooting depth, a smaller crown, and a smaller sapwood area for water transport than a PFT with a small β. A similar tradeoff has been used previously to quantify the costs and benefits associated with growing thick bark (Lawes et al., 2013) and accounts for the observed increased bark investment only in trees growing in fire-prone areas (Charles-Dominique et al., 2017; Pellegrini et al., 2017a). This approach

probably exaggerates the cost of bark because the relative cost of a tree growing outer bark for the purpose of stem insulation may be substantially less than the cost of growing xylem. Thus we performed an additional set of simulations with no carbon cost associated with tree bark investment to capture the end members of the spectrum of carbon tradeoff associated with bark investment strategy.

The advantage of having thicker bark is incorporated through our fire survivorship function. This function

prescribes that trees with thicker bark are more likely to survive a fire event than trees with thinner bark. Thus, large trees



and trees with a large β at a given height are more likely to survive than small trees and trees with a small β. Survivorship is also dependent on the amount of grass biomass present. This grass dependence comports with observations that an abundance of grass biomass results in higher ecosystem flammability and hotter fires that cause increased tree mortality (Hoffmann et al., 2012b). These ideas are incorporated in the following equation based on Pellegrini et al. (2016a) derived from data in Hoffmann et al. (2009):

$$survivorship = \begin{cases} min(1, 0.618 * bt + 0.0383) & \text{for } g_b > 25 \\ min(1, 0.602 * bt + 0.1484) & \text{for } g_b \leq 25 \end{cases}. \tag{2}$$

In Equation (2), *survivorship* is the tree survivorship fraction of a given tree size class and functional guild, $g_b$ is grass biomass (in g C m$^{-2}$), and 25 is the threshold for increased fire intensity based on grass biomass curves of savanna flammability from Hoffmann et al. (2012b). In the model, fire intensity is based only on grass biomass present and grass survivorship after fire is always zero. However, grass is able to reseed by continuous seed deposition from adjacent unburnt patches. We used the updated model to assess the joint effects of precipitation, fire disturbance, and bark investment strategy on (1) tree demography and AGB and (2) tree-grass coexistence.

## 2.2 Simulations

We conducted two classes of single-grid cell simulations. We first evaluated our model performance against observed datasets within the Cerrado region of Brazil at different fire frequencies. We then performed model experiments to assess the influence of bark investment strategy on vegetation carbon and tree-grass coexistence across a rainfall and fire gradient in the Cerrado. Our model experiments included two sets of simulations. First, we ran a control simulation that included C$_4$ grass and our updated savanna and forest PFTs, but no investment in bark or fire survival benefit (*bt*=0). Second, we ran simulations including C$_4$ grass and our updated savanna and forest tree PFTs with a fire survival strategy and allometric tradeoffs based on bark investment. This scenario assumes that bark and xylem have the same construction cost per unit volume. Given that bark construction cost may be lower than xylem construction costs, we also ran simulations including C$_4$ grass and our updated savanna and forest tree PFTs with a fire survival strategy but no cost associated with bark production. Excluding this bark-growth tradeoff did not change the overall results. We include these simulations with fire survival strategy but no bark investment tradeoff for comparison in the Supplementary Material.

In our simulations, ecosystems were spun up from tree seedlings initialized at a density of 1.0 seedling m$^{-2}$. C$_4$ grass was able to seed in at a rate of 0.001 kg C month$^{-1}$ m$^{-2}$. Simulations were forced with 0.5°, 3-hourly meteorology from the Princeton Global Forcing dataset (Sheffield et al., 2006). To isolate the effects of precipitation and fire frequency, all simulations were forced with a constant 370 ppm atmospheric CO$_2$ and meteorology looped over the same 20-year period (1980-2000). This twenty-year period is long enough to ensure a majority of the years are not anomalous, but short enough to reduce the effects of trends in climate with global change. Median fire frequency was prescribed at the beginning of a





simulation and fire interval varied stochastically within ± 3 years of the user input value. Fire disturbance occurred at most once per year and the fraction of the landscape that burned was kept constant at 30% during a given fire event. Thus, there

were no climate feedbacks on fire, and grass biomass only affected fire intensity (i.e. tree fire survival probability), but not the frequency or burned area fraction. Windthrow disturbance affected 1% of the landscape each year, resulting in 100% mortality of trees taller than 5 m and 20% mortality of trees smaller than 5 m.

### 2.2.1 Model Evaluation

We evaluated performance of the updated model using ecosystem measurements from study sites located in IBGE and the adjacent JBB Ecological Reserves within the Cerrado region of Brazil at approximately 15.95° S and 47.85° W. In these reserves, mean annual temperature is ~22.5 °C and mean annual precipitation (MAP) totals ~1460 mm with a distinct dry season from May to September. From the 1910s until reserve formation in the 1970s, the whole landscape was subject to frequent fire at ~2 year intervals (Pellegrini et al., 2014). Since reserve formation, the initiation of fire management strategies

has provided a landscape with diverse disturbance histories: fire return interval ranges from a few years in savannas to half a century in some adjacent forest patches (Pellegrini et al., 2014).

At the tree level, we compared annual diameter increments in the simulation with bark and without fire disturbance to annual diameter increments of twelve paired savanna and forest species measured over the years 2006-2007 (Rossatto et al., 2009). We performed a 35-year model spin up from seedlings in accordance with the disturbance history and age

structure of the site (Rossatto et al., 2009) and then examined the range of average annual diameter increments of our savanna and forest PFTs over a 20-year period to the range of observed annual diameter increments. In our simulations, the dbh size classes included in the calculation for the savanna PFT ranged from 5-9 cm with a mean of ~7 cm and from 6-10 cm with a mean of ~8 cm for the forest PFT in accordance with the Rossatto et al. (2009) observations.

Next, we assessed the ability of the model to predict observed tree size class distributions and measurements of

AGB at sites with different fire frequencies to evaluate if the updated model with bark could more accurately capture ecosystem-level carbon dynamics and tree size abundance in response to fire compared to the control model without bark. We compared model simulations to inventories of trees with dbh > 2 cm along 200-300 m$^2$ transects made in July 2012 within the IBGE and JBB Ecological Reserves. Total AGB was calculated from diameter inventories using the allometric relations from Xu et al. (2016) and accounting for bark carbon investment assuming β = 0.077. We performed a 35-year

model spin up in accordance with the disturbance history of the reserves after 1970 (Pellegrini et al., 2014) and then compared the range in simulated AGB over the next 10-year period in a simulation with bark fully included and a control simulation without bark investment to observed AGB estimates and tree size class distributions. In our simulations, our high fire frequency scenario was forced with a fire return interval that ranged from 2-8 years, our intermediate fire frequency scenario was forced with a return interval that ranged from 9-15 years, and our low fire frequency scenario was forced with a





return interval that ranged from 57-63 years, in accordance with the disturbance regimes of Pellegrini et al. (2014) plots 3, 4-5, and 6, respectively.

### 2.2.2 Model Experiments

To assess the outcome of including bark investment as a fire survival strategy on carbon vulnerability and tree-grass

coexistence, we conducted experiments along a rainfall gradient within the Cerrado, one with a low MAP of ~820 mm, one with an intermediate MAP of ~1150 mm, and one with a high MAP of ~1660 mm. At each location we included simulations with bark investment and a growth tradeoff, and with no bark at four forced fire return intervals ranging from 1-6, 5-11, 9-15, and 22-28 years. Additionally, we included a simulation with no fire. For each simulation we performed a 100-year model spin up from seedlings and then analysed the subsequent 20-year average total AGB, AGB by tree size, and tree

crown area fraction (Fig. 1). Tree crown area fraction ranged from zero, corresponding to open grassland, to one, corresponding to closed canopy forest and were calculated by summing the crown area of each cohort (up to a maximum crown area of one). Savanna regions were defined as regions having a crown area fraction of 0.2-0.8.

### 3 Results

**3.1 Agreement between emergent tree growth in simulations and observations**

Simulated growth rates for both the savanna and forest PFTs fell well within the observed range for Cerrado species when the allometric growth tradeoff with bark thickness was included (Fig. S1). The median growth rate for the savanna PFT was < 0.1% different from the observed median savanna species growth rate (Fig. S1a). Additionally, both observed and simulated savanna species had similar minimum diameter increments. However, the model did not capture the upper limit of

faster growing savanna species. The model overestimated the median growth rate for the forest species by ~23%, but was able to capture a wide range of the variability in growth rates seen in the observations (Fig. S1b). Further, the interquartile range of observed and simulated diameter increments overlapped broadly.

Including a bark investment strategy in the model resulted in trees becoming more fire resistant with increasing size, which reproduced the observed size class distribution in the data. For higher fire frequencies, simulations where bark

was represented as a fire survival strategy better reproduced the observed maximum tree size compared to the model without bark (Fig. 2). Further, at high fire frequencies, simulations with bark had the smallest percent difference in predictions of the median size class (23.0% compared to 25.9% for the simulation without bark) (Fig.1a). At intermediate fire frequencies, simulations with bark also had the smallest percent difference in predictions of the median size class (27.0% compared to 51.7% for the simulation without bark) (Fig. 2b). For low fire frequencies, including a bark investment strategy did not

improve predictions of maximum tree size, however a better prediction for median tree size was achieved (38.6% compared



to 44.1% for the simulation without bark) (Fig. 2c). Consequently, and as expected, the importance of bark investment for capturing ecological dynamics was greatest in frequently burning environments.

Size-specific survivorship affected predictions of ecosystem AGB under different fire frequencies. At high fire frequencies, simulations with a bark investment strategy captured the observed AGB within its predicted range and had the lower percent error between simulated and observed mean AGBs (an overestimation by 36.2% compared to an underestimation by 38.8% for the simulation without bark) (Fig. 3a). Under intermediate fire frequencies, simulations with a bark investment strategy overestimated mean AGB by 0.8%, and the observed AGB was fully within the simulated interquartile range, whereas the simulation without bark underestimated mean AGB by 20.2% and did not capture the observed AGB within the range of predicted values (Fig. 3b). At low frequency fire, the simulation without bark predicted the observed AGB marginally more accurately than the model with a bark investment strategy (percent errors of +4% and -6.7%, respectively) (Fig. 3c).

### 3.2 Bark investment decreases carbon losses at high fire frequencies

We found substantial differences in the fraction of AGB present in different tree size classes between the original model without bark and the updated model with a bark investment strategy, but these differences depended on fire frequency and precipitation. The impact of including bark thickness increased with MAP and fire frequency, likely because the higher potential growth rates allowed for trees to grow larger faster but the increased fire frequency restricted the growth of species that did not invest in thick bark (Figs. 4,S2). When fire was frequent, the model without the bark investment strategy predicted a large contribution of small and intermediate-sized trees to biomass whereas the model version with bark investment strategy predicted that virtually all biomass was in the larger tree size classes (Fig. 4). These results were robust regardless of the cost of bark investment to tree growth (see Section 2.1) (Fig. S3). The greater proportion of biomass in large trees was due to the low probability of mortality during a fire because of the insulating capacity of bark and the relationship between tree size and bark thickness (eq.1). As would be expected, the difference in tree size distributions between the model without bark and the model with a bark investment strategy decreased substantially when fires were eliminated (Fig. 4b,d,f).

We also found that tree size distributions were largely unaffected by MAP in the presence of frequent fire in both simulations with a bark investment strategy (Figs. 4a,c,e,S3); only minor impacts were found at intermediate fire frequency and high MAP (Fig. S2c). In contrast, small size classes comprised a substantial fraction of AGB were prevalent for high fire frequency simulations using the original model without a bark investment strategy, particularly at low MAP (Fig. 4a,c,e).

Incorporating bark thickness decreased predicted carbon losses with increasing fire frequency (Fig. 5) because larger, thick-barked trees made up the majority of AGB (Fig. 4) and had a very low probability of mortality during a fire. When a bark investment strategy was included, fire caused almost no reduction in biomass at the wettest site (1660 mm yr$^{-1}$ MAP), (Figs. 5a,S4a). In simulations with no bark investment strategy, trees were highly vulnerable to fire, so burning at ~3-





year intervals resulted in a 73% reduction in biomass relative to when fire was excluded (Fig. 5b). However, the effect of

including a bark investment strategy on reducing carbon losses decreased as MAP decreased (i.e., the difference in simulated carbon loss between scenarios without and with bark investment strategy was lowest in the driest site). For example, at an intermediate MAP (1150 mm yr⁻¹), a 3-year fire return interval caused a 36% reduction in AGB in the model with a bark investment strategy and an 81% reduction in AGB using the control model without bark investment, relative to ecosystems without fire. At the driest site, a 3-year fire return interval caused a 70% reduction in AGB in simulations with a bark

investment strategy and a 76% reduction in AGB for the simulations without bark, relative to ecosystems without fire.

There was a strong interaction between precipitation, fire, and bark investment strategy. When no bark investment strategy was included, both fire frequency and precipitation exerted an equivalently strong control on total AGB, and the range in AGB after 100 years of growth increased substantially with increasing precipitation, but strongly depended on fire frequency (Fig. 5b). However, when bark investment was included as a fire survival strategy, MAP exerted a much stronger

control than fire on the total AGB (Figs. 5a,S4a), indicating the important role of water availability regulating growth when species become fire resistant.

### 3.3 Environment and biological traits jointly affect simulated tree-grass coexistence

Both fire frequency and precipitation were important in maintaining tree-grass coexistence and thus in controlling

the distribution of grasslands, savannas, and tropical forests (Fig. 6). At high fire frequencies and low precipitation, we simulated grasslands with minimal tree cover regardless of the model scheme (Figs. 6a,c,S5a,c). However, simulations with bark investment as a fire survival strategy expanded conditions under which there was 20-80% tree-grass coexistence. At fire return intervals of 1-6 years and intermediate MAP (1150 mm yr⁻¹) and at less frequent fire return intervals (9-15 years) and intermediate MAP (1150 mm yr⁻¹), bark investment mitigated either complete tree loss or complete grass loss (Figs. 6,S5).

Thus, including bark investment as a fire survival strategy moderated the transition between grassland and forest at intermediate MAP under varying fire frequencies.

### 4 Discussion

We documented that tree bark investment strategy interacts with precipitation and fire frequency to determine both

(i) the stability of ecosystem carbon to fire and (ii) the coexistence of grasses and trees, illustrating that species traits, in addition to climate and fire, are critical for the stability of savanna and forest biomes. Bark investment strategy increased the stability of the carbon stock in large trees, which decreased ecosystem carbon losses with increased fire frequency under a range of precipitation conditions. Investment in bark was especially important in wetter savannas and forests, illustrating that the distribution of functional traits is fundamental to the resilience of wet forests to increased fire and changing rainfall

regimes.





### 4.1 Implications for carbon resistance

Our simulations illustrate that tree bark thickness as a fire survival mechanism substantially decreases fire-driven carbon losses, but the magnitude of the effect depends on precipitation regime (Fig. 5a-b). We found that bark investment

was particularly important at reducing carbon losses at higher fire frequencies in locations with a high MAP (1660 mm). This is because trees had ample water availability, enabling them to grow rapidly and decreasing the probability of fire mortality to near zero. In contrast, the effect of bark investment on reducing carbon losses diminished at the lowest MAP and highest fire frequency because trees were not able to grow rapidly and accumulate thick enough bark to escape fire mortality. Taken together, these results suggest that current models that do not account for bark investment strategies may underpredict

Neotropical carbon resistance to fire in both savannas and forests. Such models would over-predict mortality of large thick-barked trees that make up a majority amount of aboveground woody biomass (Slik et al., 2013; Hanan et al., 2008). However, further work understanding the spatial distribution of tree species and their corresponding bark investment strategies is also critical (Pellegrini et al., 2017a; Rosell, 2016; Pausas, 2015; Dantas et al., 2013) because observations show that even large trees in rainforests have thin enough bark that they suffer substantial fire mortality (Uhl and Kauffman,

1990), although the forests with high mortality have precipitation values much higher than the maximum we consider (1660 mm).

### 4.2 Implications for tree-grass coexistence

Capturing savanna distributions globally has long been difficult for vegetation models, which over-predicted the

extent of either tropical forests or grasslands (Hickler et al., 2006; Cramer et al., 2001; Bonan et al., 2003; Hely et al., 2006; Schaphoff et al., 2006; Sato et al., 2007). A number of recent studies have focused on this issue: the adaptive dynamic global vegetation model (aDGVM) was able to capture savanna extent in Africa by including (i) trees with a higher fire mortality rate in small tree size classes, (ii) regenerative tree resprouting after fire events, and (iii) grass as super individuals (Scheiter and Higgins, 2009). The individual-based Populations-Order-Physiology model also included size-dependent tree mortality

and was able to reproduce key vegetation structure and function along a rainfall and fire gradient in Australia (Haverd et al., 2013). Studies by Baudena et al. (2015) and Lasslop et al. (2016) have proposed several key mechanisms for capturing savannas in models: (1) water limitation on tree growth, (2) competition for water between grasses and trees, and (3) a grass-fire feedback.

The results from this study show a strong dependence of tree growth on precipitation and fire frequency, supporting

observations (Pellegrini et al., 2017b; Higgins et al., 2007) and other modeling studies (Baudena et al., 2015). However, we found that including bark investment as a fire survival mechanism broadened the range of climate and fire conditions under which savannas occur by reducing the range of conditions leading to either complete tree loss or complete grass loss. This is

due to the shift towards fewer, larger trees that store relatively more AGB per unit crown area than smaller trees, but are also resilient to frequent fire, resulting in ecosystems with intermediate crown cover in a wider range of precipitation and fire regimes. Thus, fire and precipitation as well as species-specific bark thickness traits have the potential to affect tree-grass coexistence, suggesting that inclusion of bark investment in models has the potential to substantially enhance our ability to accurately project changes in the tropical carbon sink with changes in fire and rainfall over the upcoming century.

**4.3 Limitations and future work**

Several avenues exist for future model improvement. Tree re-sprouting after fire has been shown to be essential in predicting the range of conditions for which tree-grass coexistence is possible (Higgins et al., 2000). Currently in our model, trees reproduce only through seedling recruitment. This has the potential to affect simulation outcomes because initial aboveground growth rates of re-sprouts are substantially higher compared to seedlings due to (a) access to belowground carbohydrate stores and (b) elimination of the need to allocate carbon for roots. Thus, re-sprouting may allow for better agreement between model predictions at higher growth rates and fire frequencies (Figs. 2,3,S1). Enabling tree re-sprouting may also stabilize savannas under frequent fire and low precipitation where all model versions currently simulate grassland (Fig. 6).

Additionally, the current fire model does not interact dynamically with climate or the nitrogen cycle. ED2 is capable of resolving nitrogen dynamics (Trugman et al., 2016) and we anticipate that coupling nitrogen and climate feedbacks to fire will be an important step in accurately modelling the carbon cycle. Despite these current limitations, the updated model accurately predicts growth, demographics, and AGB. We believe that these results can provide important insight into tree-grass coexistence and carbon resistance with changing fire frequency in the Neotropics with global change.

**5 Conclusions**

In conclusion, our results highlight that carbon storage in tropical savannas and forests depends not only on changing environmental drivers, but also on tree fire survival strategy. Thus, we can improve projections of the tropical carbon sink with global climate change by increasing our understanding of the distribution of bark investment and incorporating this knowledge of bark investment into future vegetation models. Further, an increased understanding of the interaction between bark investment strategy and environmental drivers promises to increase our ability to project the distribution of savanna in regions previously simulated as grasslands or forests by reducing the range of conditions leading to either complete tree loss or complete grass loss.

**Code availability:** Model codes are included as an online supplement



**Competing interests:** The authors declare that they have no conflict of interest

**Author contributions**

A.T.T., A.F.A.P., D.M., W.A.H. designed the research. A.T.T performed numerical experiments. A.T.T. and A.F.A.P. analysed the data. A.F.A.P. and W.A.H. contributed field data. A.T.T drafted the paper and all authors contributed to writing

of the manuscript.

**Acknowledgments**

The authors gratefully acknowledge support from the Program in Latin American Studies at Princeton University (A.F.A.P), the Princeton Environmental Institute Walbridge Fund (A.T.T), the National Science Foundation Graduate Research

Fellowship (A.T.T and A.F.A.P), the National Science Foundation Award 1151102. (D.M.), and US Department of Energy, Office of Science, Office of Biological and Environmental Research, Terrestrial Ecosystem Science (TES) Program under award number DE-SC0014363 (D.M.).







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





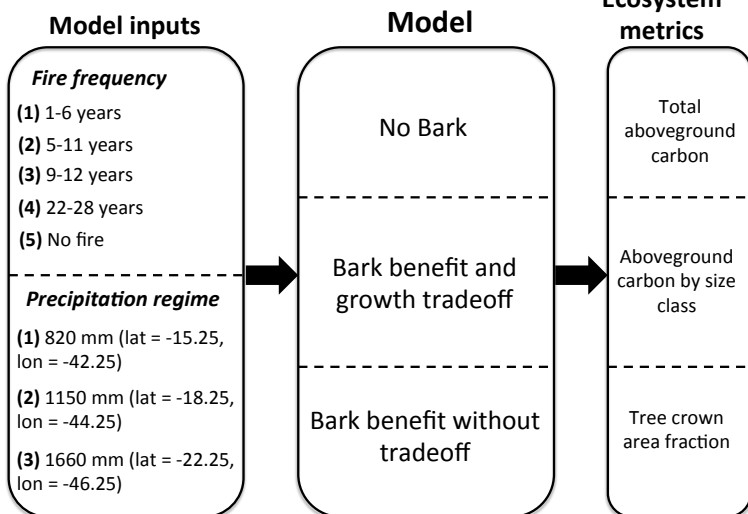

**Figure 1.** Schematic of model experiments along a rainfall gradient. Varied model inputs include fire frequency and precipitation regime (associated with a particular location in the Cerrado region of South America). Different model versions were used to understand the effect of including bark as a fire survival strategy on model outputs of various ecosystem metrics including aboveground woody carbon, tree size class distribution, and tree crown area fraction.



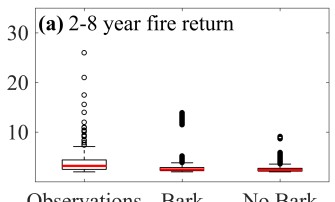

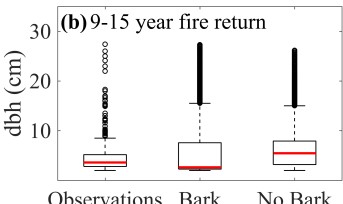

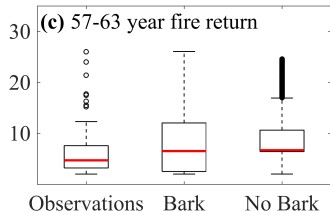

**Figure 2.** Tree fire survival strategy impacts ecosystem tree demography with fire disturbance. Observed and model-simulated tree diameter at breast height (dbh) size class distributions under **(a)** high, **(b)** intermediate, and **(c)** low frequency fire regimes. Model simulations included a 35-year model spin up in accordance with the disturbance history of the observations (Pellegrini et al., 2014). Simulated tree size was compared over the subsequent 10-year period for the model with bark and the model without bark. The red line denotes the median tree size, the black box denotes the interquartile range, and the dotted error bars denote ± 2.7 standard deviations.





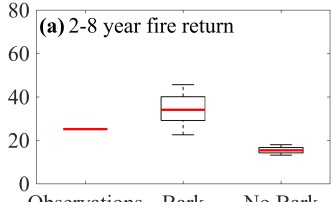

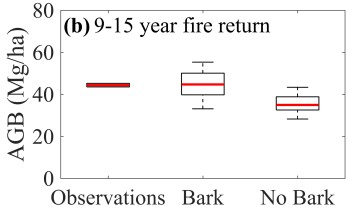

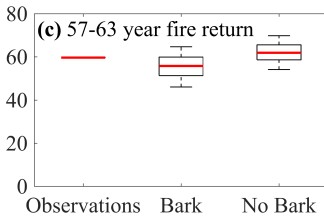


**Figure 3.** Tree fire survival strategy impacts aboveground woody carbon with fire disturbance. Observed and model-simulated aboveground woody biomass (AGB) under **(a)** high, **(b)** intermediate, and **(c)** low frequency fire regimes. Model simulations included a 35-year model spin up in accordance with the disturbance history of the observations (Pellegrini et al., 2014). Simulated AGB was compared over the subsequent 10-year period for the model with bark and the model without bark. The red line denotes the median AGB, 525    the black box denotes the interquartile range, and the dotted error bars denote ± 2.7 standard deviations.





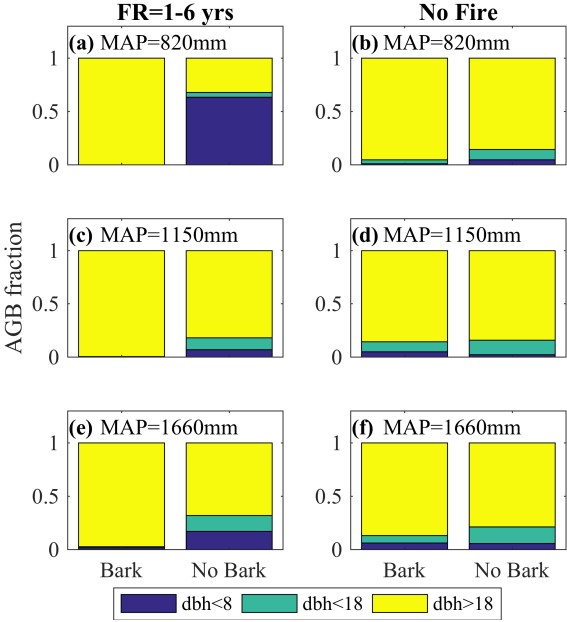

**Figure 4.** Bark fire survival strategy increases the fraction of aboveground woody biomass in large trees, particularly with frequent fire. Model-simulated fraction of aboveground woody biomass (AGB) present in different tree diameter at breast height (dbh in cm) size classes at low **(a-b)**, intermediate **(c-d)**, and high **(e-f)** MAP for a high frequency (FR) fire regime **(a,c,e)**, and a no fire disturbance simulation **(b,d,f)**. Simulations were initialized with tropical tree and $C_4$ grass plant functional types and included a 100-year model spin up from seedlings. The subsequent 20-year average AGB fraction by tree size class is shown for the model with bark and the model without bark. See Fig. S2 for size class distributions associated with intermediate fire frequency.





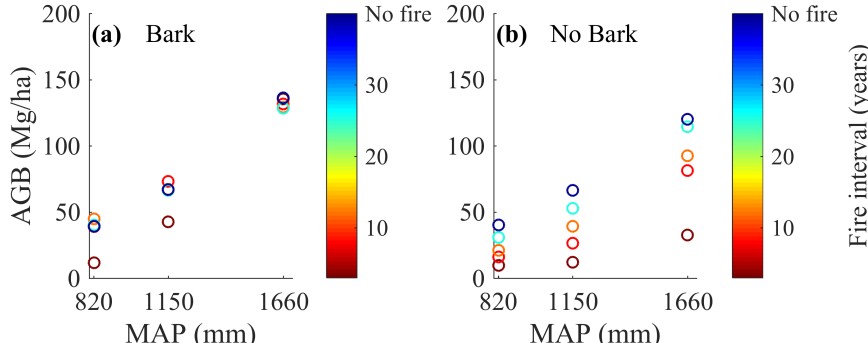

**Figure 5.** Bark fire survival strategy buffers aboveground woody biomass loss with frequent fire, particularly at high MAP. Model-simulated total aboveground woody carbon (AGB) at different MAP and forced fire regimes for the model with bark **(a)** and the model with no bark **(b)**. Simulations were initialized with tropical tree and $C_4$ grass plant functional types and included a 100-year model spin up from seedlings. The subsequent 20-year average AGB for each disturbance and precipitation regime is shown.



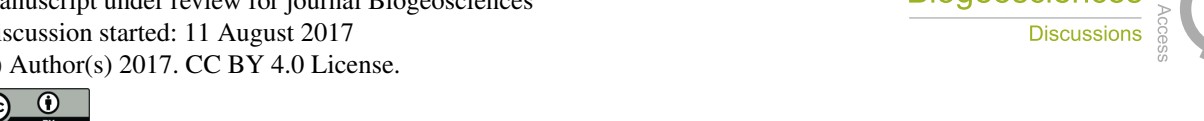

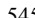

545

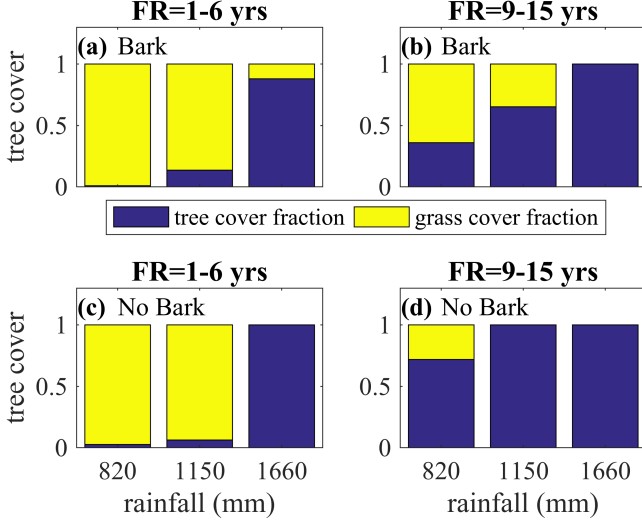

**Figure 6.** Bark investment can broaden the range of climate and fire conditions under which savannas occur by reducing the range of conditions leading to either complete tree loss or complete grass loss. Model-simulated tree cover fraction present at different levels of mean annual precipitation (in mm) for the model with bark **(a-b)** and the model without bark **(c-d)** for fire frequencies (FR) of 1-6 years **(a,c)** and 9-15 years **(b,d)**. Simulations were initialized with tropical tree and $C_4$ grass plant functional types and included a 100-year model spin up from seedlings. The subsequent 20-year average tree cover fraction is shown.



**Tables**

555

**Table 1** | Model evaluations at IBGE Ecological Reserve

| Metric | Fire frequency | Data source |
|---|---|---|
| **(1)** Annual diameter increments of twelve paired savanna and forest species | **(1)** No fire | Rossatto et al. (2009) |
| **(2)** Inventory tree size class | **(1)** 2-8, **(2)** 9-15, and **(3)** 57-63 years | Pellegrini et al. (2014) plots 3, 4-5, and 6 |
| **(3)** Inventory aboveground biomass | **(1)** 2-8, **(2)** 9-15, and **(3)** 57-63 years | Pellegrini et al. (2014) plots 3, 4-5, and 6 |