# Peer review of "Sensitivity of woody carbon stocks to bark investment strategy in Neotropical savannas and forests"

_Biogeosciences, 2017_

## Referee Comment (RC1) · Anonymous Referee #1 · 25 Sep 2017

General comments

In the age of anthropogenically forced climate change, developing a comprehensively predictive Earth System Model is an extremely high scientific priority. Trugman et al is a highly relevant study that aims to assess hypotheses regarding the trade-off between fire adaptation in the form of bark thickness and growth and survivorship. The hypotheses evaluated in this study are clearly articulated in the Introduction and described in the Methods such that replicating this experiment would be fairly straightforward. ED2 is an appropriate tool and the Methods are, in general, appropriately designed for exploring the hypotheses of this study. Importantly, the source code for the model, which

documents the exact model structure and parameter values used in this study, is included with the Supplement Information, thus making the model itself transparent and the simulations reproducible. This study makes two important contributions: 1) (model development) it demonstrates the importance of including a fire-adapted trait in terrestrial biosphere models and 2) (ecological understanding) it illustrates the potential role bark thickness has in modulating coexistence of different life forms (grasses vs. trees) and aboveground C-stocks in fire-regulated ecosystems. I do, however, recommend that, before this manuscript is accepted for publication, the authors address the concerns listed below and make clarifications in the text where necessary.

Specific comments

1. The Methods need to be clearer (refer to lines 96-100, 144-146) about the two new tropical PFTs not being simulated simultaneously. As written, the Methods left me with the impression that both PFTs were being run simultaneously and that direct selection for one strategy over the other was part of the experiment (line 144-146). I had to read the namelist (ex. ED2IN_fire_control_MAP1450_FR3_pen1) to verify that only the savanna tree or forest tree (apparently PFT #26 for both?) was being run with the C4 grass (PFT #1?) in each simulation. This needs to be clear so that the reader can fully understand how to interpret the ecological significance of the results.

2. I am not strongly convinced by the ecological importance of the results presented in Figures 2 and 3. In several panels it is difficult to visually discern that the "bark" performed better than the "no bark" or what the differences, where present, actually mean. For example, in all of the panels the IQR of the "no-bark" is tighter than the "bark", and hence, is in better agreement with the IQR of the observations. Or, for Fig 3a, I would not necessarily consider the "bark" to do a better job "capturing" the observations relative to the "no-bark" just because the observations fell within the broad error range of the bark-model predictions (line 224). Benchmarking new model capability is important, and I think these are obvious choices as benchmarks. However, I am not sure what these benchmarks add to the central argument (and scientific contribution) of this

paper (e.g. as I take it, the central argument is: Abstract lines 23-25, Introduction line 6, Figures 5 and 6, Results lines 275-276, Discussion line 280, 316-317, Conclusions lines 343-346). Consider moving Figures 2 and 3 to the Supplemental. Even if model performance against the benchmarks in Figures 2 and 3 is weak, as I would contend, I do not view this as a major problem for this paper because the emergent pattern in Figure 6 supports the "bark" hypothesis that this study seeks to evaluate.

3. In replacement of Figures 2 and 3 in the main text, it would be nice to see a figure of how the demographic rates that give rise to Figures 5 and 6 differ between "bark" and "no-bark". Demography is the fundamental feature of the ED2 hypothesis, as explained in Moorcroft et al. 2001. Therefore, it would be informative to see how the internal dynamics of the model that are central to its hypothesis (i.e. predictions of demographic rates—growth, recruitment, mortality) are modified by inclusion of the bark strategy. In essence, this is the interesting ecology that ED2 and this study have great potential to inform. Figure S1 is the only figure of demographic rates and it is an informative starting point, but by itself, it does not close-the-loop for explaining the patterns in Figures 5 and 6. It seems to me that a mortality figure would also be useful for helping to explain the interaction between the survivorship hypothesis (Eqn2) and the different growth rates.

4. The meaning of the variation in Figures 2 and 3 is unclear. I do not understand how time factors into these distributions. Presumably, the observations are based on census data that was collected at one point in time, July 2012 (line 182). How was the model output sampled to generate the error estimate? If the errors are different—one spatial (or across individuals), one temporal—then how do we compare the errors? Please clarify in the caption, text or both.

5. I suggest a stylistic revision regarding verb tense for sentences describing model predictions. Since the model is a hypothesis, the verb tense should be in the present when speaking about model predictions. For example, at line 240 it says "predicted", but it should read "predicts". "The hypothesis predicts. . ." = "The model predicts. . ."

6. I suggest that the final sentence of the Abstract focus on the scientific contribution to ecology and not on model development. Model development speaks to a subset of ecologists. Since the Abstract may be the only thing many people read, the concluding sentence should be about the most important scientific contribution and speak to the broadest community you are trying to reach.

7. Line 46. "comes at a growth cost: thicker-bark...grows more slowly...". This sentence is written as if this the carbon-cost trade-off is conclusively known, but it is not; it is a hypothesis. Indeed it is a hypothesis evaluated in this study with the no-cost bark PFT. Consider rephrasing.

8. Line 72. "We then tested whether two hypotheses were consistent with model simulations." This does not read correctly to me. The model is the hypothesis and the simulations reflect the hypothesis. So, it is hard for me to rationalize how a hypothesis is consistent with itself. Consider revising.

9. Section 2.1. The source of the model parameter values needs to be cited. The only obvious reference for parameter values is for the brevideciduous PFT at line 97. For example, it is not clear what version of ED2 is the basis for the photosynthesis parameterization used in this study.

10. Line 168. Please provide the definition of the dry season used in this study (<100 mm/month?). It would also be helpful to know what fraction of annual precipitation falls during the dry season.

11. Line 210. Could tuning model parameters unrelated to bark thickness correct the overestimation of the forest species?

12. Line 235. "likely". This is speculative. This can be known with this model. Addressing item 3 above should make this known.

13. In the model code, the namelist needs to have a description of what PFT 26 actually is. There are several PFTs listed up to number 17 in the namelist; but after that, it is not

clear what 26 is or what the other tropical PFTs (numbers 2-4) are. The PFTs above 17 are given some notation in the source code (e.g. in ed_params.f90), but it is still not clear what exactly that notation means. A little more clarification about the notation throughout the code would be helpful.

Technical comments

1. Line 18. "ED2" Define acronym.

2. Line 215. "Fig1a". Wrong reference?

3. Line 248. "comprised a substantial fraction of AGB were prevalent..." This does not read smoothly to me. Consider revising.

---

## Referee Comment (RC2) · Anonymous Referee #2 · 26 Sep 2017

Bark thickness can protect trees against fire damage and mortality. This feature is particularly important for survival in fire driven ecosystems such as savannas. Most dynamic vegetation models do, however, not consider tree plant functional types with variable bark thickness and hence with different levels of resistance against fire. In this study, bark thickness is considered in the ED2 vegetation model to describe fire survivorship and it is explored how the introduction of more fire tolerant trees influences vegetation dynamics in neotropical forests and savannas. The authors argue that including fire tolerant trees improves agreement with empirical data and that it can increase the areas where savannas can occur.

The manuscript investigates and interesting and important question and it may contribute to our understanding of the distribution of savannas and how we can better model savannas. The manuscript is generally well written and formulates hypotheses that are then tested by model simulations.

I have some comments concerning the results. Generally I think that statistical test should be conducted to quantify agreement with data and differences between model runs. I am for example not convinced that in Fig 2a, the "Bark" simulations are better than the "No Bark" simulations. Maximum dbh of with bark simulations is higher than in the no bark simulations but differences in means are not visible in the panel. Histograms for the dbh or height distribution might be more illustrative than box plots.

Fig. 4 suggests that there are more or less no small trees in with bark simulations while we often find many small trees in savannas due to the high re-sprouting rates. This biomass distribution suggests that re-sprouting and recruitment are not possible (re-sprouting is not included in the model and is identified as a limitation in the discussion) but I assume that it would strongly influence small tree numbers. I wonder how stable this vegetation state is: if simulations were continued and all tall trees die, would the simulated vegetation converge to a grassland without any trees because regrowth is not possible?

Most analyses investigate vegetation in response to variable frequency while timing or intensity are not considered. Yet, these variables strongly influence vegetation responses to fire.

It is stated in I. 60 that "DGVMs are still unable to fully capture global savanna extent". It would be very interesting to see how the updated model version influences the savanna distribution at larger spatial scales both in comparison to the original model version and to other DGVMs. I think this is not the scope of this study, nonetheless this point could be mentioned in the discussion.

Further comments:

I. 53: "Slower growth rates result in a population of smaller trees with relatively thinner bark" I would argue that bark thickness is not relevant for small trees anyway because they are in the flame zone and damaged by each fire. The capacity to regrow after fire might be more important. Bark thickness is mainly important for tall trees that managed to escape flame height.

I. 62: suggest to reword to "carbon storage in the tropics"

I. 106: Please check table, I can't find definition of beta in Table 1.

I. 195: I suggest to make clear that the tree MAP levels are sites along the rainfall gradient.

I. 217: Fig 2a instead of 1a?

I. 521: Fig 3 shows biomass but the caption says "woody carbon". Please check text for consistency.

Fig 5: I suggest to replace the current color legend with a legend showing color and the associated fire return interval. Also I suggest to use a consistent notation: fire interval or fire frequency.

---

## Author Comment (AC1) · 11 Oct 2017

→ We thank Referee 1 for the helpful feedback and suggestions on our manuscript. Our responses are listed below. We invite further dialogue if anything is unclear or if more explanation is required.

General comments In the age of anthropogenically forced climate change, developing a comprehensively predictive Earth System Model is an extremely high scientific priority. Trugman et al is a highly relevant study that aims to assess hypotheses regarding the trade-off between fire adaptation in the form of bark thickness and growth and survivorship. The hypotheses evaluated in this study are clearly articulated in the Introduction and described in the Methods such that replicating this experiment would be fairly straightforward. ED2 is an appropriate tool and the Methods are, in general, appropriately designed for exploring the hypotheses of this study. Importantly, the source code for the model, which documents the exact model structure and parameter values used in this study, is included with the Supplement Information, thus making the model itself transparent and the simulations reproducible. This study makes two important contributions: 1) (model development) it demonstrates the importance of including a fire-adapted trait in terrestrial biosphere models and 2) (ecological understanding) it illustrates the potential role bark thickness has in modulating coexistence of different life forms (grasses vs. trees) and aboveground C-stocks in fire-regulated ecosystems. I do, however, recommend that, before this manuscript is accepted for publication, the authors address the concerns listed below and make clarifications in the text where necessary.

→ We thank Reviewer 1 for the positive feedback highlighting our manuscript's contributions to both model development and ecological understanding. Below we list how we will address Reviewer 1's concerns and the corresponding changes and clarifications that we will make when the discussion period has ended.

Specific comments

1. The Methods need to be clearer (refer to lines 96-100, 144-146) about the two new tropical PFTs not being simulated simultaneously. As written, the Methods left me with the impression that both PFTs were being run simultaneously and that direct selection for one strategy over the other was part of the experiment (line 144-146). I had to read the namelist (ex. ED2IN_fire_control_MAP1450_FR3_pen1) to verify that only the savanna tree or forest tree (apparently PFT #26 for both?) was being run with the C4 grass (PFT #1?) in each simulation. This needs to be clear so that the reader can fully understand how to interpret the ecological significance of the results.

→ We apologize for the confusion. The savanna, forest, and C4 grass PFTs were

all run simultaneously for all model experiments (3 PFTs total). We will clean up the namelist files in the source code and include a readme file in the run folder with instructions on how to generate the namelist for the different model experiments. We will also clarify that 3 PFTs were used in the methods so that the reader can fully understand how to interpret the ecological significance of the results.

2. I am not strongly convinced by the ecological importance of the results presented in Figures 2 and 3. In several panels it is difficult to visually discern that the "bark" performed better than the "no bark" or what the differences, where present, actually mean. For example, in all of the panels the IQR of the "no-bark" is tighter than the "bark", and hence, is in better agreement with the IQR of the observations. Or, for Fig 3a, I would not necessarily consider the "bark" to do a better job "capturing" the observations relative to the "no-bark" just because the observations fell within the broad error range of the bark-model predictions (line 224). Benchmarking new model capability is important, and I think these are obvious choices as benchmarks. However, I am not sure what these benchmarks add to the central argument (and scientific contribution) of this paper (e.g. as I take it, the central argument is: Abstract lines 23-25, Introduction line 6, Figures 5 and 6, Results lines 275-276, Discussion line 280, 316-317, Conclusions lines 343-346). Consider moving Figures 2 and 3 to the Supplemental. Even if model performance against the benchmarks in Figures 2 and 3 is weak, as I would contend, I do not view this as a major problem for this paper because the emergent pattern in Figure 6 supports the "bark" hypothesis that this study seeks to evaluate.

→ Your point is valid and Reviewer 2 shared similar concerns. We will experiment with plotting model evaluations as histograms to see if this better illustrates the difference in the distribution skew for bark and no bark simulations. We will also consider moving Figures 2-3 to the supplemental and replacing them with suggestions from your point 3 (below), although we are somewhat hesitant to remove all model validations from the main text because benchmarking new model capabilities is an important component to any modeling paper.

3. In replacement of Figures 2 and 3 in the main text, it would be nice to see a figure of how the demographic rates that give rise to Figures 5 and 6 differ between "bark" and "no-bark". Demography is the fundamental feature of the ED2 hypothesis, as explained in Moorcroft et al. 2001. Therefore, it would be informative to see how the internal dynamics of the model that are central to its hypothesis (i.e. predictions of demographic rates of growth, recruitment, mortality) are modified by inclusion of the bark strategy. In essence, this is the interesting ecology that ED2 and this study have great potential to inform. Figure S1 is the only figure of demographic rates and it is an informative starting point, but by itself, it does not close-the-loop for explaining the patterns in Figures 5 and 6. It seems to me that a mortality figure would also be useful for helping to explain the interaction between the survivorship hypothesis (Eqn2) and the different growth rates.

→ Thank you for this suggestion, it is very insightful. Adding an additional figure to help unpack the different demographic processes in the model will improve the manuscript significantly. We will experiment with illustrating the mortality rates by size classes in the different model versions at different fire frequencies. Additionally, we also include fire intensity/probability of mortality dependent on grass biomass (line 131) in the model. We will further experiment with illustrating how variable fire intensity affects size-specific tree mortality.

4. The meaning of the variation in Figures 2 and 3 is unclear. I do not understand how time factors into these distributions. Presumably, the observations are based on census data that was collected at one point in time, July 2012 (line 182). How was the model output sampled to generate the error estimate? If the errors are one spatial (or across individuals), one temporal then how do we compare the errors? Please clarify in the caption, text or both.

→ The range in the model output represents the range in simulated AGB or tree size over a 10 year period following a 35-year model spin up (lines 184-186). Variation in the observations is due to variation between plot biomass or in variation across individuals

within plots. Although the errors are not analogous, we intended to represent the range in model output resulting from growth and variable meteorological forcing over the 10-year time period. We will clarify this in the text and the captions.

5. I suggest a stylistic revision regarding verb tense for sentences describing model predictions. Since the model is a hypothesis, the verb tense should be in the present when speaking about model predictions. For example, at line 240 it says "predicted", but it should read "predicts". "The hypothesis predicts. . ." = "The model predicts. . ."

→ We will revise the verb tense describing the model predictions according to your suggestion.

6. I suggest that the final sentence of the Abstract focus on the scientific contribution to ecology and not on model development. Model development speaks to a subset of ecologists. Since the Abstract may be the only thing many people read, the concluding sentence should be about the most important scientific contribution and speak to the broadest community you are trying to reach.

→ We will revise the abstract's final sentence to focus on the ecological significance of this work, possible incorporating results from suggestion #3 looking at how the demographic rates that give rise to different biomass retention and tree-grass coexistence.

7. Line 46. "comes at a growth cost: thicker-bark. . .grows more slowly. . .". This sentence is written as if this the carbon-cost trade-off is conclusively known, but it is not; it is a hypothesis. Indeed it is a hypothesis evaluated in this study with the no-cost bark PFT. Consider rephrasing.

→ We will rephrase this sentence to highlight the uncertainty in the cost of bark investment.

8. Line 72. "We then tested whether two hypotheses were consistent with model simulations." This does not read correctly to me. The model is the hypothesis and the simulations reflect the hypothesis. So, it is hard for me to rationalize how a hypothesis

is consistent with itself. Consider revising.

→ We will rephrase this sentence.

9. Section 2.1. The source of the model parameter values needs to be cited. The only obvious reference for parameter values is for the brevideciduous PFT at line 97. For example, it is not clear what version of ED2 is the basis for the photosynthesis parameterization used in this study.

→ We cite Xu et al (2016) on line 97, but we will clarify that this model version is the base version that we use.

10. Line 168. Please provide the definition of the dry season used in this study (<100 mm/month?). It would also be helpful to know what fraction of annual precipitation falls during the dry season.

→ We will better characterize dry season absolute precipitation and precipitation fraction in the text.

11. Line 210. Could tuning model parameters unrelated to bark thickness correct the overestimation of the forest species?

→ Tuning model parameters such as the maximum rate of carboxylation could correct for the overestimation of forest species. In this study we considered the hypothesis that the carbon tradeoff on bark investment alone was responsible for the differences in growth rate between savanna and forest species.

12. Line 235. "likely". This is speculative. This can be known with this model. Addressing item 3 above should make this known.

→ We will modify this based on results from item 3.

13. In the model code, the namelist needs to have a description of what PFT 26 actually is. There are several PFTs listed up to number 17 in the namelist; but after that, it is not clear what 26 is or what the other tropical PFTs (numbers 2-4) are. The PFTs above

17 are given some notation in the source code (e.g. in ed_params.f90), but it is still not clear what exactly that notation means. A little more clarification about the notation throughout the code would be helpful.

→ We will clean up the source code and include descriptions of the full list of PFTs.

Technical comments

1. Line 18. "ED2" Define acronym.

2. Line 215. "Fig1a". Wrong reference?

3. Line 248. "comprised a substantial fraction of AGB were prevalent. . ." This does not read smoothly to me. Consider revising.

→ We will correct the above technical errors in the manuscript.

---

## Author Comment (AC2) · 11 Oct 2017

→ We thank Referee 2 for the helpful feedback and suggestions on our manuscript. Our responses are listed below along with corresponding changes that we will make to the text once the discussion period ends. We invite further dialogue if anything is unclear or if more explanation is required.

Bark thickness can protect trees against fire damage and mortality. This feature is particularly important for survival in fire driven ecosystems such as savannas. Most dynamic vegetation models do, however, not consider tree plant functional types with variable bark thickness and hence with different levels of resistance against fire. In

this study, bark thickness is considered in the ED2 vegetation model to describe fire survivorship and it is explored how the introduction of more fire tolerant trees influences vegetation dynamics in neotropical forests and savannas. The authors argue that including fire tolerant trees improves agreement with empirical data and that it can increase the areas where savannas can occur.

The manuscript investigates and interesting and important question and it may contribute to our understanding of the distribution of savannas and how we can better model savannas. The manuscript is generally well written and formulates hypotheses that are then tested by model simulations.

I have some comments concerning the results. Generally I think that statistical test should be conducted to quantify agreement with data and differences between model runs. I am for example not convinced that in Fig 2a, the "Bark" simulations are better than the "No Bark" simulations. Maximum dbh of with bark simulations is higher than in the no bark simulations but differences in means are not visible in the panel. Histograms for the dbh or height distribution might be more illustrative than box plots.

→ We will experiment with plotting model evaluations as histograms and quantify the skew of distributions for each bark and no bark simulations to illustrate the differences in model performance.

Fig. 4 suggests that there are more or less no small trees in with bark simulations while we often find many small trees in savannas due to the high re-sprouting rates. This biomass distribution suggests that re-sprouting and recruitment are not possible (resprouting is not included in the model and is identified as a limitation in the discussion) but I assume that it would strongly influence small tree numbers. I wonder how stable this vegetation state is: if simulations were continued and all tall trees die, would the simulated vegetation converge to a grassland without any trees because regrowth is not possible?

→ First, as a clarification, the minimum tree size in all figures should read

2cm<dbh<8cm, as we excluded trees smaller than 2cm dbh in our modeling results because trees smaller than 2cm were not surveyed in our observational comparisons. We apologize for not making this clear and will do so in the revision. The biomass fraction of small trees is not illustrative of the number frequency of small trees, as small tree biomass comprises a very small fraction compared to the large trees, so it is not necessarily the case that small trees are not present in the bark simulations. We will add additional figures showing the number count of trees by size class to better illustrate the demographic processes based on this confusion and a suggestion from Reviewer 1.

Most analyses investigate vegetation in response to variable frequency while timing or intensity are not considered. Yet, these variables strongly influence vegetation responses to fire.

→ We include variable fire intensity/probability of mortality dependent on grass biomass (line 131) in the model simulations. We will further experiment with including results for how variable fire intensity affects tree mortality.

It is stated in l. 60 that "DGVMs are still unable to fully capture global savanna extent". It would be very interesting to see how the updated model version influences the savanna distribution at larger spatial scales both in comparison to the original model version and to other DGVMs. I think this is not the scope of this study, nonetheless this point could be mentioned in the discussion.

→ We will update the manuscript to include in the discussion about capturing savanna extent on larger spatial scales.

Further comments:

l. 53: "Slower growth rates result in a population of smaller trees with relatively thinner bark" I would argue that bark thickness is not relevant for small trees anyway because they are in the flame zone and damaged by each fire. The capacity to regrow after fire

might be more important. Bark thickness is mainly important for tall trees that managed to escape flame height.

→ Bark thickness could be relevant to smaller trees in that it could slow growth and prolonged the period during which trees are more susceptible to fire, depending on environmental conditions. Additionally, bark thickness might influence the threshold at which a tree could be considered large enough to no longer be susceptible to fire.

l. 62: suggest to reword to "carbon storage in the tropics" l. 106:

→ We will reword according to your suggestion.

Please check table, I can't find definition of beta in Table 1. → We will remove the reference to Table 1, as beta is defined in the main text in line 106-110. We apologize for this stray reference from a previous draft.

l. 195: I suggest to make clear that the tree MAP levels are sites along the rainfall gradient.

→ We will clarify this in the main text.

l. 217: Fig 2a instead of 1a?

→ We will update the reference to 2a.

l. 521: Fig 3 shows biomass but the caption says "woody carbon". Please check text for consistency.

→ We will update for consistency.

Fig 5: I suggest to replace the current color legend with a legend showing color and the associated fire return interval. Also I suggest to use a consistent notation: fire interval or fire frequency.

→ We will consistently use fire frequency in the text.